# Development and validation of a multiplex real-time qPCR assay using GMP-grade reagents for leprosy diagnosis

**Fernanda Saloum de Neves Manta**[1☉], **Thiago Jacomasso**[2☉], **Rita de Cássia Pontello Rampazzo**[2], **Suelen Justo Maria Moreira**[1], **Najua M. Zahra**[2], **Stewart T. Cole**[3,4], **Charlotte Avanzi**[3,5], **Thyago Leal-Calvo**[1], **Sidra Ezidio Gonçalves Vasconcellos**[6], **Phillip Suffys**[6], **Marcelo Ribeiro-Alves**[7], **Marco Aurelio Krieger**[2,8], **Alexandre Dias Tavares Costa**[2,8‡*], **Milton Ozório Moraes**[1‡*]

1 Laboratório de Hanseníase, Instituto Oswaldo Cruz, FIOCRUZ, Rio de Janeiro, Brazil, 2 Instituto de Biologia Molecular do Paraná, FIOCRUZ, Curitiba, Brazil, 3 Global Health Institute, École Polytechnique Fédérale de Lausanne, Lausanne, Switzerland, 4 Institut Pasteur, Paris, France, 5 Department of Microbiology, Immunology and Pathology, Mycobacteria Research Laboratories, Colorado State University, Fort Collins, Colorado, United States of America, 6 Laboratório de Biologia Molecular Aplicada a Micobactérias, Instituto Oswaldo Cruz, FIOCRUZ, Rio de Janeiro, Brazil, 7 Instituto Nacional de Infectologia Evandro Chagas, Rio de Janeiro, Brazil, 8 Laboratório de Ciências e Tecnologias Aplicadas à Saúde (LaCTAS), Instituto Carlos Chagas, Fundação Oswaldo Cruz/FIOCRUZ, Curitiba, Brazil

☉ These authors contributed equally to this work.
‡ ADTC and MOM also contributed equally to this work.
* alexandre.costa@fiocruz.br (ADTC); milton.moraes@fiocruz.br (MOM)

**Data Availability Statement:** All relevant data are within the manuscript and its Supporting Information files.

## Abstract

Leprosy is a chronic dermato-neurological disease caused by *Mycobacterium leprae*, an obligate intracellular bacterium. Timely detection is a challenge in leprosy diagnosis, relying on clinical examination and trained health professionals. Furthermore, adequate care and transmission control depend on early and reliable pathogen detection. Here, we describe a qPCR test for routine diagnosis of leprosy-suspected patients. The reaction simultaneously amplifies two specific *Mycobacterium leprae* targets (16S rRNA and RLEP), and the human 18S rRNA gene as internal control. The limit of detection was estimated to be 2.29 copies of the *M. leprae* genome. Analytical specificity was evaluated using a panel of 20 other skin pathogenic microorganisms and *Mycobacteria*, showing no cross-reactivity. Intra- and inter-operator $C_p$ variation was evaluated using dilution curves of *M. leprae* DNA or a synthetic gene, and no significant difference was observed between three operators in two different laboratories. The multiplex assay was evaluated using 97 patient samples with clinical and histopathological leprosy confirmation, displaying high diagnostic sensitivity (91%) and specificity (100%). Validation tests in an independent panel of 50 samples confirmed sensitivity and specificity of 97% and 98%, respectively. Importantly, assay performance remained stable for at least five months. Our results show that the newly developed multiplex qPCR effectively and specifically detects *M. leprae* DNA in skin samples, contributing to an efficient diagnosis that expedites the appropriate treatment.

**Funding:** This work was funded by a grants from: Banco Nacional de Desenvolvimento Econômico e Social (BNDES), contract no. 15.2.0473.1 (Operation #4.816.864) to MAK, and Novartis Foundation and Leprosy Research Initiative (LRI; 703.15.45), Foundation for Research Support of the State of Rio de Janeiro (FAPERJ;E-26/203.053/2016), Brazilian National Council for Scientific and Technological Development (CNPq; 421852/2017-2018), Brazilian Coordination for Improvement of Higher Education Personnel (CAPES), and by the National Fund for Health/Brazilian Ministry of Health (MS/SCTIE/DECIT; 404277/2012-8 and TED 145/2018) to MOM. The funders had no participation in the present study's design, data collection, analysis, interpretation, or writing of the report and decision to submit for publication.

**Competing interests:** I have read the journal's policy and the authors of this manuscript have the following competing interests: TJ and RdCPR were employed by IBMP at the time the manuscript was accepted. IBMP has commercial interest in the reactions described in the manuscript. IBMP had no participation in the present study's design, data collection, analysis, interpretation, or writing of the report and decision to submit for publication.

## Author summary

Leprosy is a chronic dermato-neurological disease caused by *Mycobacterium leprae*, an obligate intracellular bacterium. Diagnosis of leprosy often relies on skin examinations for clinical signs, bacilli staining from skin smears and invasive skin biopsies. However, the spectrum of clinical manifestations and, often, low bacilli numbers can hinder accurate diagnosis. Timely detection is a challenge in leprosy diagnosis, relying on clinical examination and requiring trained health professionals. Proper intervention for adequate care and transmission control depends on early and reliable pathogen detection. Quantitative PCR methods for detecting bacterial DNA are more sensitive and could aid in differentially diagnosing leprosy from other dermatological conditions. In this work, we present a new multiplex PCR that was assessed for quality control standards, and the data indicate that the assay is stable and reproducible. The results presented here are the basis of a novel and robust tool with potential to increase the accuracy of leprosy diagnosis in routine or reference laboratories.

## Introduction

Leprosy is a neglected infectious disease that still represents a public health issue [1] with more than 200,000 cases every year worldwide. Diagnosis is generally late and, although a specific and effective treatment is available, it is likely that transmission occurs before the patient is diagnosed and adequately treated, thus contributing to sustained transmission. The high number of young patients (under 15 years old) and patients with disabilities due to the advanced stage of the disease, confirms this hypothesis [1]. Furthermore, clinical forms vary to a great extent, from localized (tuberculoid) to disseminated (lepromatous) forms, making diagnosis difficult. Evidence suggests that early diagnosis could prevent transmission and help epidemiological control [2].

Methods such as bacterial index detection by microscopy and histopathological examination have been the main complementary tools for the diagnosis of leprosy [2–4]. Classical bacteriological methods cannot confirm leprosy since *M. leprae* does not grow *in vitro*. In addition, there is no reliable marker to estimate the risk of disease progression [5,6]. In this regard, the sequencing of *M. leprae* genome [7] was a milestone towards the improvement of direct *M. leprae* detection, leading not only to better characterization of genomic targets unique to *M. leprae* strains but also to an extensive comparison of different mycobacteria.

At the time when the first sequences became available, the polymerase chain reaction (PCR) technique was laborious and very expensive, averting its universal application. However, as PCR was further developed, it became more affordable, versatile and reliable, with fully automated systems becoming commercially available from different companies [8–10]. For tuberculosis, routine tests using PCR reduced the turnaround time, allowing same day treatment initialization, which might impact resistance prevalence [11–13]. Cost-effective nucleic acid detection assays are relatively widespread, but assays for some neglected diseases are still missing. In leprosy, the situation is even more difficult due to reduced and late investments directed to diagnostic tests [14].

In the last few years, many studies have been carried out using the PCR technique to detect *M. leprae* DNA in clinical specimens. PCR has been used especially under challenging diagnoses such as equivocal paucibacillary [4,15–18] or monitoring household contacts [19,20]. In this context, several different targets have been described in an attempt to establish the most sensitive and specific assay [16,20–28]. However, most of the PCR protocols were developed,

evaluated, and validated using reagents or tests produced without good manufacturing practices (GMP). Also, most of the studies enroll only leprosy patients and do not recruit patients with other common dermatological diseases that are differential diagnosis to leprosy. Thus, the development and validation of an assay over different laboratories has become a necessity.

Here, we present the development and validation of a multiplex real-time PCR assay aiming to standardize the leprosy molecular diagnostic assay. The protocol was designed to simultaneously detect two *M. leprae* targets (16S rRNA and RLEP genes), previously used in several studies [4,16,19,26,29], and one mammalian target (18S rRNA gene), that serves as reaction control [30]. Cross-reactivity was evaluated using DNA from 20 related mycobacterial and other skin pathogenic species, and no match was found. The new assay was validated using 97 skin biopsies and an independent panel enrolling 50 samples retrieved from patients previously characterized by clinical examination and histopathology, showed high sensitivity and specificity. The new multiplex PCR was also assessed for quality control standards and the data indicate that the assay is stable and reproducible. The results presented here are the basis of a novel and robust tool with potential to increase the accuracy of leprosy diagnosis in routine or reference laboratories.

## Material and methods

### Ethics statement

The Ethics Committee of the Oswaldo Cruz Foundation approved this study (CAAE: 38053314.2.0000.5248, number: 976.330-10/03/2015). Written informed consent was obtained from all patients. If participant was a minor, written formal consent was obtained from the parent/guardian.

### Clinical samples

Leprosy patients were enrolled at the Leprosy clinic from the Oswaldo Cruz Foundation in the city of Rio de Janeiro, Brazil. Skin biopsies were collected using a 6 mm punch and stored in 70% ethanol at -20˚C until processing. The samples were included according to patient enrollment consecutively and are representative of the period between 2010 and 2018.

Ninety-seven samples (53 skin biopsies from leprosy patients and 44 skin biopsies from patients with other skin diseases) were used for qPCR tests. Clinical and demographic characteristics of all patients are shown in Table 1.

Leprosy patients were defined according to the clinical, bacteriological, and histopathological Ridley-Jopling (R&J) classification and the operational classification in multibacillary (MB) or paucibacillary (PB) forms according to the WHO [31]. Leprosy or other dermatological diseases (ODD) patients were treated according to their respective condition. Leprosy paucibacillary (PB) or multibacillary (MB) patients were treated according to the Ministry of Health recommendations, while ODD patients were treated accordingly for each specific disease.

### Replication Study

To validate the conditions and analysis parameters established with the clinical samples from Oswaldo Cruz's Leprosy Clinic, we tested a distinct collection of 50 skin biopsy samples that were also obtained at the Leprosy Clinic. The second set of samples was sent to the Global Health Institute, École Polytechnique Fédérale de Lausanne, Switzerland, where DNA samples were extracted and characterized by conventional PCR according to a previously published protocol [32]. Then, purified DNA was sent back to the Leprosy Clinic at Oswaldo Cruz Foundation,

**Table 1. Clinical and demographic characteristics of the leprosy and other dermatological disease cases.**

| Characteristics | Types | 1st panel | | 2nd panel | |
|---|---|---|---|---|---|
| | | Leprosy group (n = 53) | ODD group (n = 44) | Leprosy group (n = 35) | ODD group (n = 15) |
| Gender | Male | 32 | 13 | 25 | 3 |
| | Female | 21 | 31 | 10 | 12 |
| Age | 1–15 | 2 | 3 | 3 | 1 |
| | 16–30 | 8 | 7 | 5 | 2 |
| | 31–45 | 14 | 6 | 11 | 2 |
| | 46–60 | 20 | 19 | 9 | 9 |
| | >60 | 9 | 9 | 7 | 1 |
| WHO classification | PB | 18 | NA | 8 | NA |
| | MB | 35 | NA | 27 | NA |
| Clinical form | I | 6 | NA | 0 | NA |
| | TT | 1 | NA | 3 | NA |
| | BT | 11 | NA | 5 | NA |
| | BB | 5 | NA | 6 | NA |
| | BL | 3 | NA | 7 | NA |
| | LL | 27 | NA | 14 | NA |
| Bacterial index | 0 | 23 | 38 | 8 | 15 |
| | 0–2 | 6 | 0 | 2 | 0 |
| | 2–4 | 11 | 0 | 13 | 0 |
| | 4–6 | 13 | 0 | 12 | 0 |

Other Dermatological Disease (ODD). Operational classifications [paucibacillary (PB) or multibacillary (MB)]. PB individuals were classified as Tuberculoid (TT), Borderline tuberculoid (BT), and Indeterminate (I). MB individuals were classified as Borderline-borderline (BB), Borderline lepromatous (BL) or Lepromatous (LL). NA: Not Applicable.

where it was blindly analyzed with the qPCR developed in the present study. After analysis, blinding was removed and the results were compared. Of these 50 samples, fifteen were from patients with other skin diseases, 27 patients had MB leprosy and eight from PB leprosy. The group presented a 1.27:1 ratio of males to females. The mean age was 44.8 (± 17.72 SD), and the range was 8–77. Details on the clinical characteristics are shown in S1 Table.

## Mycobacterial isolates samples

*M. leprae* Thai-53 purified from athymic BALB/c (*nu/nu)* mouse footpads was kindly provided by Dr. Patricia Rosa at the Lauro de Souza Lima Institute, Bauru, São Paulo, Brazil. Purified DNA from *M. leprae* was used as positive control and in analytical sensitivity studies. DNA from 21 mycobacterial samples were used for the analytical specificity study. *L. amazonensis* and *L. braziliensis* was kindly provided by Dr Elisa Cupolillo by the Laboratório de Pesquisa em Leishmaniose (IOC- FIOCRUZ), and *M. avium*, *M. gordonae*, *M. manteni*, *M. africanum* subtype I, *M. africanum* subtype II, *M. bovis*, *M. bovis* (BCG), *M. canettii*, *M. fortuitum*, *M. gordonae*, *M. intracellulare*, *M. kansasii*, *M. microti*, *M. pinnipedii*, *M. simiae*, and *M. tuberculosis* DNA was extracted at the Laboratório de Biologia Molecular Aplicada a Micobactérias (IOC--FIOCRUZ) as published elsewhere [33].

## Synthetic DNA

The synthetic DNA (gBlock) was purchased from Integrated DNA Technologies (IDT) and consists of a double-stranded DNA containing the sequences of the three genomic targets

(RLEP, 16S rRNA, and 18S rRNA) (S1 Text). The lyophilized DNA was reconstituted to 10 ng/μL (corresponding to $1.83 \times 10^9$ copies per reaction) in TE pH 8.0, following the supplier's protocol.

## DNA extraction

DNA extraction from the biopsies was carried out using the DNeasy Blood and Tissue extraction kit (Qiagen, Germany). The total extracted DNA was quantified with NanoDrop (Thermo-Fisher Scientific, Waltham, MA, USA) and stored at -20˚C. *M. leprae* DNA from nude mice footpad was purified using TRIzol reagent (Life Technologies, Carlsbad, California) following the manufacturers' instructions, as previously described [3]. DNA used in the replication study were extracted using QIAmp UCP Pathogen Mini kit (Qiagen GmbH, Hilden, Germany).

## Standard curve and 95% limit of detection ($LoD_{95\%}$) assessment

The standard curve was used for determination of the limit of detection and assay stability. A series of 10-fold dilutions was prepared from either *M. leprae* or synthetic DNA, using DNA purified from human blood obtained from healthy donors as a sample matrix. The dilution series used for the standard curve and the $LoD_{95\%}$ determination spans concentrations from 0.5 fg/reaction to 5 ng/reaction of purified *M. leprae* DNA (approximately $1.4 \times 10^{-1}$ to $1.4 \times 10^6$ genome-equivalents/reaction, considering the M. leprae genome size to be 3.3 Mbp), and $1.83 \times 10^0$ to $1.83 \times 10^7$ copies/reaction (equivalent to 0.5 ag/reaction and 5 pg/reaction, respectively) of synthetic DNA.

## Quantitative PCR (real-time PCR assays)

A multiplex real-time qPCR assay targeting simultaneously two *M. leprae* regions and an internal reference human sequence was developed. The primers and hydrolysis probes were designed to detect regions from RLEP and 16S rRNA genes [29] from *M. leprae*, and the human 18S rRNA [30] (Table 2). According to BLAST and *in silico* PCRs, the RLEP primers are specific to *M. leprae* without any complete (whole oligonucleotide) identical matching to *M. lepromatosis* or any other mycobacteria. However, 16S rRNA primers completely match *M. lepromatosis* genome sequences, except for a single base mismatch in the probe sequence. Reactions were performed on an ABI7500 Standard instrument (Thermo-Fisher Scientific, Waltham, MA, USA), using Multiplex PCR Mastermix (IBMP/FIOCRUZ PR, Curitiba, Brazil). For each reaction, 5 μL of DNA solution was added for a 25 μL final volume. Reaction mixtures were prepared in triplicates and amplified at 95˚C for 10 min, and 45 cycles of 95˚C for 15 sec and 60˚C for 1 min. All reactions included a positive control (mouse foot-pad *M.*

**Table 2. Sequences, concentration, and fluorophores of the oligonucleotides contained in the multiplex qPCR assay.**

| Target | Sequences | Final concentration | Fluorophore |
|---|---|---|---|
| **16S rRNA** | **Forward:** 5´-GCATGTCTTGTGGTGGAAAGC- 3´<br>**Reverse:** 5´-CACCCCACCAACAAGCTGAT- 3´<br>**Probe:** 5´-CATCCTGCACCGCA-3´ | 0.5 μM<br>0.5 μM<br>0.2 μM | FAM |
| **RLEP** | **Forward:** 5´-GCAGCAGTATCGTGTTAGTGAA-3´<br>**Reverse:** 5´-CGCTAGAAGGTTGCCGTAT-3´<br>**Probe:** 5´CGCCGACGGCCGGATCATCGA-3´ | 0.2 μM<br>0.2 μM<br>0.1 μM | VIC |
| **18s rRNA** | **Forward:** 5´-GAAACTGCGAATGGCTCATTAAATCA- 3´<br>**Reverse:** 5´-CCCGTCGGCATGTATTAGCTCT-3´<br>**Probe:** 5´GGAGCGAGCGACCAAAGGAACCA-3´ | 0.06 μM<br>0.06 μM<br>0.03 μM | CY5 |

*leprae* DNA and/or high-bacterial load lepromatous leprosy patient purified DNA), and water as a non-template control (NTC; PCR reaction without any template DNA).

## Stability

The stability of the new multiplex qPCR was evaluated the synthetic DNA template diluted in TE to the concentrations of approximately $2 \times 10^8$, $2 \times 10^7$, $2 \times 10^6$, $2 \times 10^6$, $2 \times 10^5$, $2 \times 10^4$, $2 \times 10^3$, $2 \times 10^2$, $2 \times 10^1$, 10, 5, and 2.5 copies per reaction.

All reagents (oligomix 25X and qPCR mix) were maintained in independent aliquots at -20°C at the Leprosy Laboratory (FIOCRUZ-RJ). Tests with the dilution series described above were repeated weekly for the first month, and then once a month for five months.

## Data analyses and statistics

Qualitative (diagnostic sensitivity and specificity, accuracy) and quantitative (intra- and inter-laboratory repeatability and reproducibility, analytical sensitivity and specificity) validation tests were performed. The 95% limit of detection ($LoD_{95\%}$) was calculated by fitting a Probit model to the estimated detection probabilities. Samples that fall under the "equivocal" category were considered negative when calculating the diagnostic sensitivity, specificity and the predictive values. These parameters were calculated according to Altman and Bland [34]. Data were processed and analyzed using customized scripts for R version 3.5.1 (downloaded from http://www.Rproject.org/).

## Results

### Analytical performance

Primers and hydrolysis probes designed to target 16S rRNA and RLEP sequences of *M. leprae* were tested in multiplexed reactions to concomitantly detect the human 18S rRNA sequence. Optimal fluorescence thresholds were chosen based on the common practice that it should be positioned on the lower half of the fluorescence accumulation curves plot from the 10-fold dilutions, crossing most if not all fluorescence signals on the exponential segment of the curve on a logarithmic scale (Fig 1). Therefore, after setting the baseline to the automatic function, fluorescence threshold values chosen for determining $C_p$ (Crossing point) values for each target were set to intercept the positive controls and avoid the negative ones, being established as follows: 0.2 for RLEP, 0.15 for 16S rRNA, and 0.16 for 18S rRNA.

The analytical 95% limit of detection ($LoD_{95\%}$) was determined from a series of tests in which DNA extracted from *M. leprae* was diluted from 5 ng to 100 ag/reaction. Fig 2 shows the fitted Probit models and the obtained $LoD_{95\%}$ for 16S rRNA and RLEP, which were experimentally determined as approximately 450 fg of DNA (ca. 126 *M. leprae* genomes) for the 16S rRNA gene and about 4.60 fg of DNA (ca. 1.3 *M. leprae* genomes) for the RLEP gene.

The developed multiplex reaction was evaluated against a collection of microorganisms to assess the specificity of the primers and probes under these conditions. The selection included several mycobacteria, as well as a few other pathogens associated with skin diseases such *Leishmaniasis* (Fig 3). We only considered any species as cross-reactive if all the technical replicates displayed amplification for at least one of the targets, which was not the case for any of the species tested. Most positive amplifications observed correspond to RLEP, which was detected in two out of three replicates in *M. fortuitum* and *M. kyroniense*. Even though some reactions presented 16S rRNA signals above the threshold, these amplifications are very uncharacteristic and are easily distinguishable from a proper amplification when compared with the positive control with 500 fg/reaction of *M. leprae*.

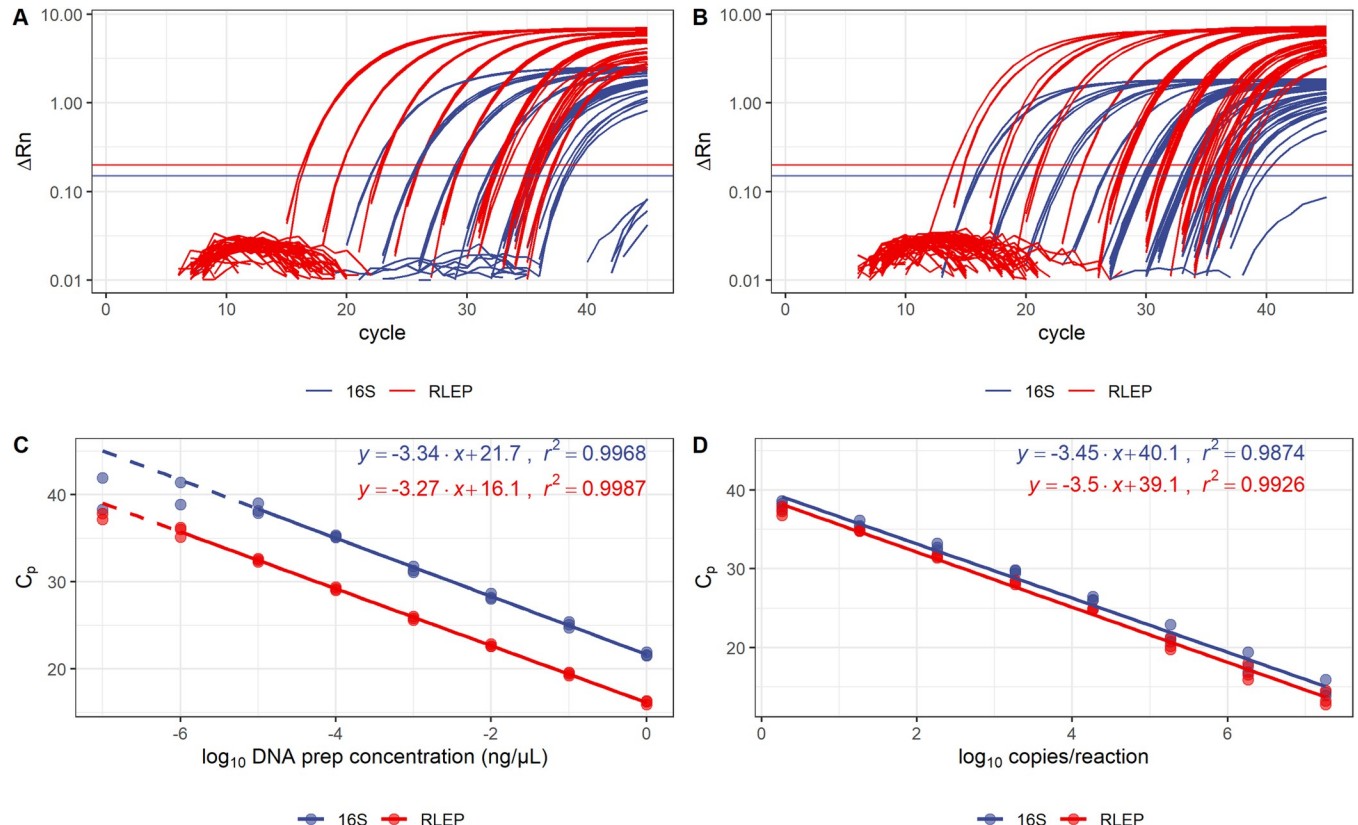

**Fig 1. Standard curves of the amplification of 16S rRNA and RLEP targets in *M. leprae* DNA and in a synthetic construct.** Panels A and C show the calibration curves obtained using *M. leprae* DNA, diluted in total DNA extracted from *M. leprae*-negative whole blood. Continuous lines show the linear range and the dashed lines are extrapolations towards the non-linear range. Efficiencies calculated from the linear ranges were 99.2% for 16S rRNA and 102.2% for RLEP, and $r^2$ were 0.9968 and 0.9987, respectively. Panels B and D show the calibration curves obtained using a synthetic gene containing one copy of each target per molecule, diluted in total DNA extracted from *M. leprae*-negative whole blood. The efficiencies were 94.9% for 16S rRNA and 93% for RLEP, and $r^2$ were 0.9874 and 0.9926, respectively.

## Repeatability and reproducibility

Three independent operators performed three replicate runs each, in consecutive days, and evaluated the repeatability and reproducibility of the multiplex reactions. For each replicate, a new dilution series for the synthetic gene was prepared from a concentrated aliquot to be used as a template. The data (S2 Table) shows that all intra-operator replicates were remarkably reproducible, with only one point (Op. 1, 16S rRNA $1.83 \times 10^2$) displaying a relative standard deviation (rRSD%) above 5%, but still well below 10%. The inter-operator variability was also very low, and the largest variation was observed for the 16S rRNA target. Nonetheless, the rRSD% was between 1.38 and 11.57 across the dilution range, which shows an excellent reproducibility for a quantitative test (see also S3 Table).

The accuracy of the determinations performed by the multiplex real-time qPCR assay was also estimated using the synthetic DNA. To evaluate the intra- and inter-repeatability (or intermediate precision) for operators, we calculated the arithmetic mean, standard deviation, and relative standard deviation percentage of three independent experiments. It is noteworthy that the detection of the human target 18S rRNA does follow the same dilution trend for the other targets because the synthetic template was not diluted in human DNA.

In summary, for both *M. leprae* targets we observed that all points showed excellent reproducibility and repeatability. As expected, detection of the human target 18S rRNA loses

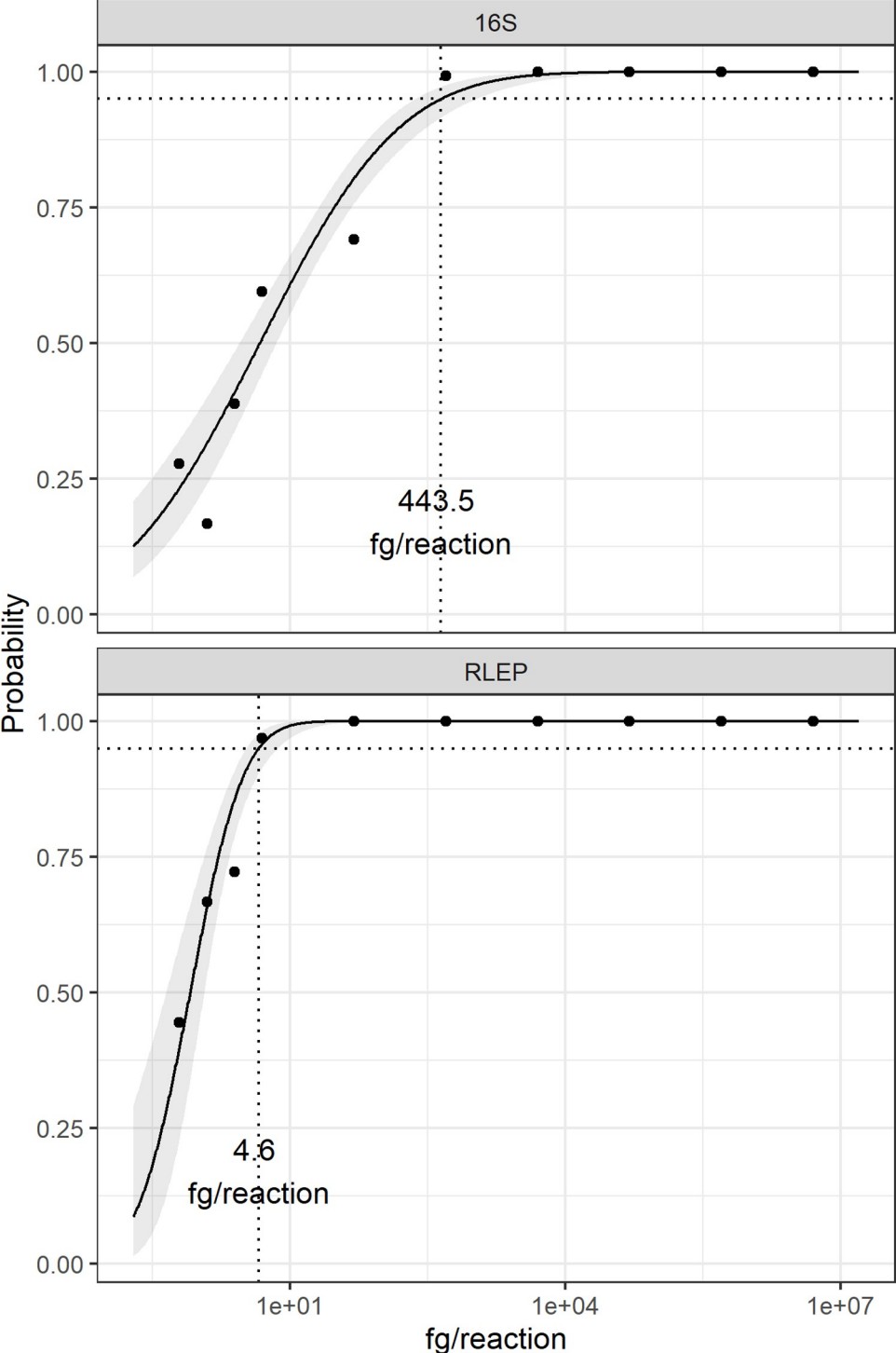

**Fig 2. Analytical 95% limit of detection (LoD$_{95\%}$) for 16S rRNA and RLEP in multiplexed qPCR.** *Mycobacterium leprae* DNA was diluted in DNA extracted from whole blood from healthy donors and tested from 5 ng to 0.5 fg/reaction. Probability of detection was calculated for 16S rRNA and RLEP (top and bottom panels, respectively) from nine independent experiments, and a Probit model was fit to the data (black lines). The gray ribbon around the model fit indicates the 95% CI on the predicted probability. Dotted lines indicate the interpolation to determine the concentration at a 95% probability. The calculated LoD$_{95\%}$ is displayed on each plot in femtograms of DNA/reaction.

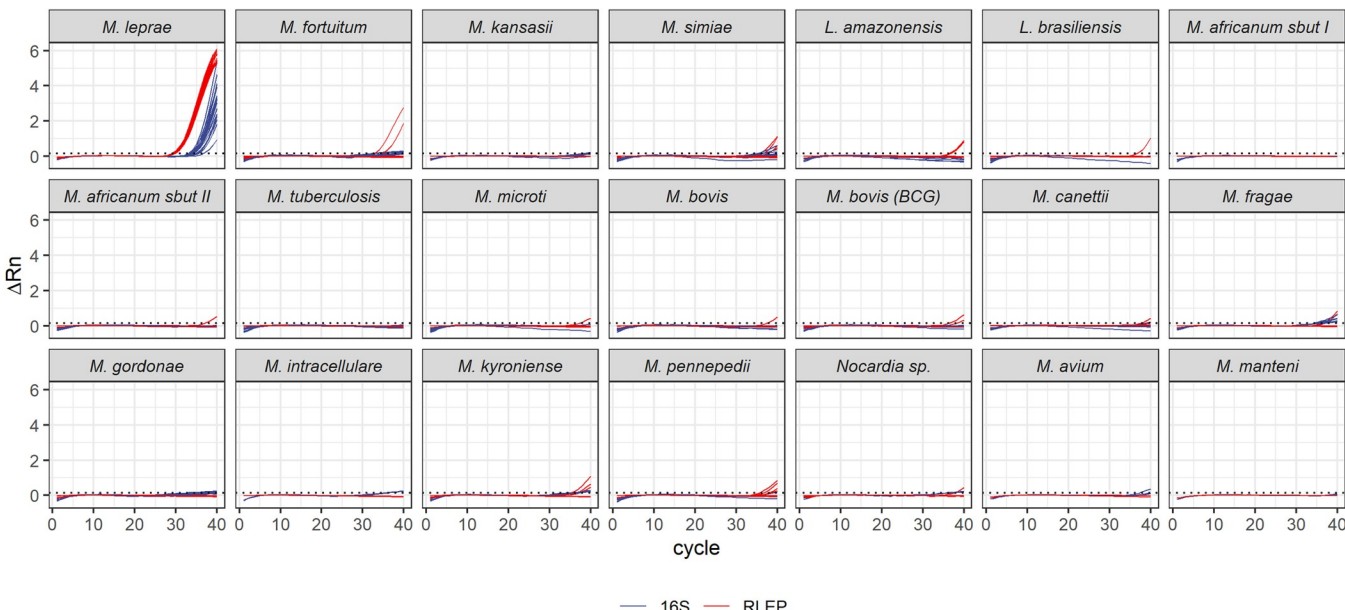

**Fig 3. Analytical specificity for the 16S rRNA and RLEP multiplexed reactions.** Extracted DNA from the indicated microorganisms (5 ng/μL each) were used in the multiplexed reactions performed in technical triplicates in two independent experiments. Results are compared to the amplification plot for 100 fg *M. leprae* DNA/μL (top-left panel). Amplification profiles are shown for each target, and each line corresponds to one individual well. The dotted lines indicate the threshold for RLEP (which is the highest of the two *M. leprae* targets, at 0.2).

reproducibility as it becomes scarce in the reaction due to the dilution factor. It is noteworthy that there is no variation in the detection of the human target 18S rRNA when *M. leprae* DNA was present in the synthetic control molecule, i.e., in a 1:1 ratio, supporting the notion that the multiplexed reactions do not interfere with each other.

## Stability

Storage stability was assessed by performing monthly evaluations of reactions with different concentrations of the synthetic DNA molecule for 5 months. Most of the data points tested varied below the established limit of three standard deviations above the average of all time points. Fig 4 shows the $C_p$ obtained for the three evaluated targets (16S rRNA, RLEP, and 18S rRNA) in representative concentrations for brevity, over a 5-month period. The test remained reliable for the entire range of concentrations tested.

## Diagnostic performance

The implemented setup involved the interrogation of two target sequences from *M. leprae* to classify clinical samples correctly while mitigating possible false positives. To evaluate the diagnostic performance, we first established optimal parameters for the analysis, considering possible cross-reactions that may occur in the laboratory routine. The $C_p$ cutoff for both targets were determined iteratively by analyzing the receiver operating characteristic (ROC) curves for each combination of cutoff cycles. S1 Fig shows the ROC curves for a subset of the best-performing combinations of cutoff values for 16S rRNA and RLEP. Data for the full range of $C_p$ cutoff combinations are listed in the S4 Table.

Based on these results the best combination of cutoff values (35.5 for 16S rRNA and 34.5 for RLEP) showed a sensitivity of 91% and specificity of 100%, positive and negative predictive values (PPV and NPV) were 100% and 90%, respectively. These values were similar between

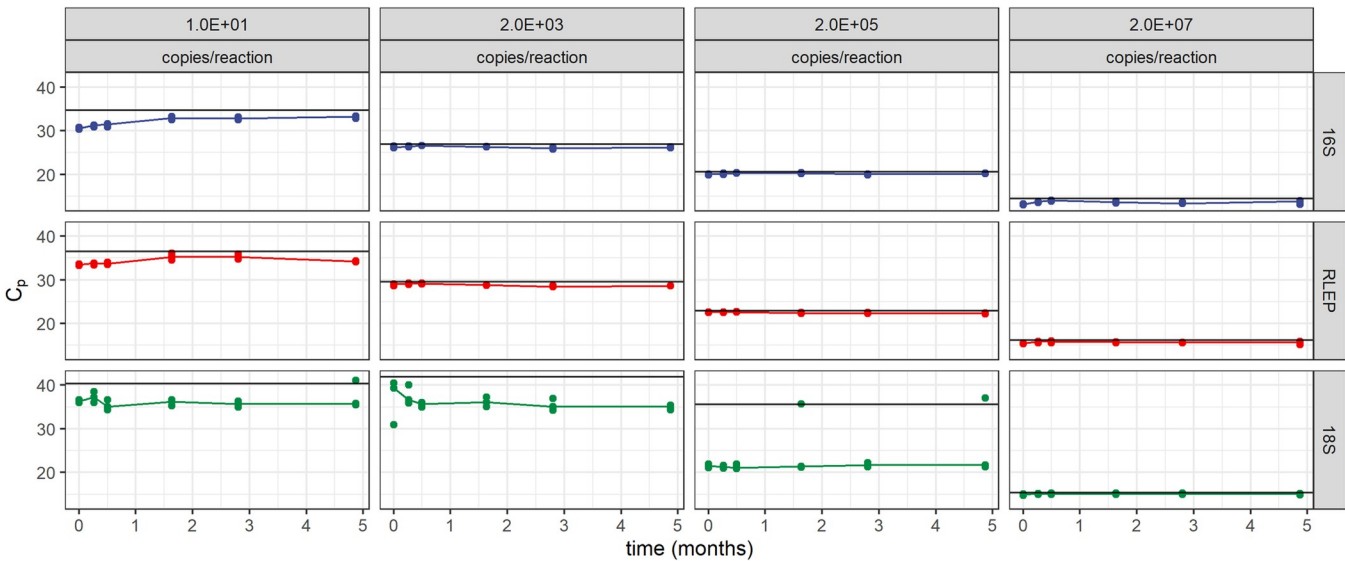

**Fig 4. Stability of the reactions over five months using synthetic DNA as a template.** Each panel shows the $C_p$ values obtained for each target (lines of panels) and for each template concentration (columns of panels) over time. Points represent one technical replicate. Black horizontal lines indicate the upper tolerance limit defined as three standard deviations above the mean $C_p$ for each template concentration.

MB and PB samples, as shown in the S5 Table. These parameters for analysis and summarized them in the decision algorithm presented in Table 3.

Next, the molecular diagnosis obtained using the new multiplex PCR, was compared to the clinical diagnosis of each sample (Fig 5 and S6 Table). Results show that the qPCR reaction and classification algorithm correctly characterized 48 of the 53 samples previously described as "Leprosy" by the clinical outcome. Of the 5 misclassified samples, one was classified as negative for *M. leprae* and four were in the "equivocal" quadrant. All the misclassified samples have a Bacterial Index of 0.

None of the 44 samples characterized as "Other skin diseases" were classified as *M. leprae*-positive by our reaction and decision algorithm. Thirty-eight of these samples were classified as "Negative" and 6 as "equivocal".

## Assay validation

Conditions established with the training cohort were tested on an independent set of samples, which were previously characterized using a distinct qPCR method described in Girma et al. [32]. The comparison between the original classification and the new results is shown in Fig 6 and S1 Table. We tested 50 samples, of which 34 were previously characterized as positive and 16 as negative.

**Table 3. Decision algorithm for classification of samples based on the data obtained with the new multiplex qPCR.**

| Results | Classification |
| --- | --- |
| 18S rRNA negative | Extraction failure (repeat extraction) |
| 18S rRNA $C_p$ between 13 and 32 | Valid reaction (proceed with classification) |
| RLEP $< 34.5$ and 16S rRNA $< 35.5$ | *M. leprae* detected |
| RLEP $< 34.5$ and 16S rRNA $\geq 35.5$ | Equivocal (mark patient for new sample collection and testing) |
| RLEP $\geq 34.5$ | *M. leprae* undetected |

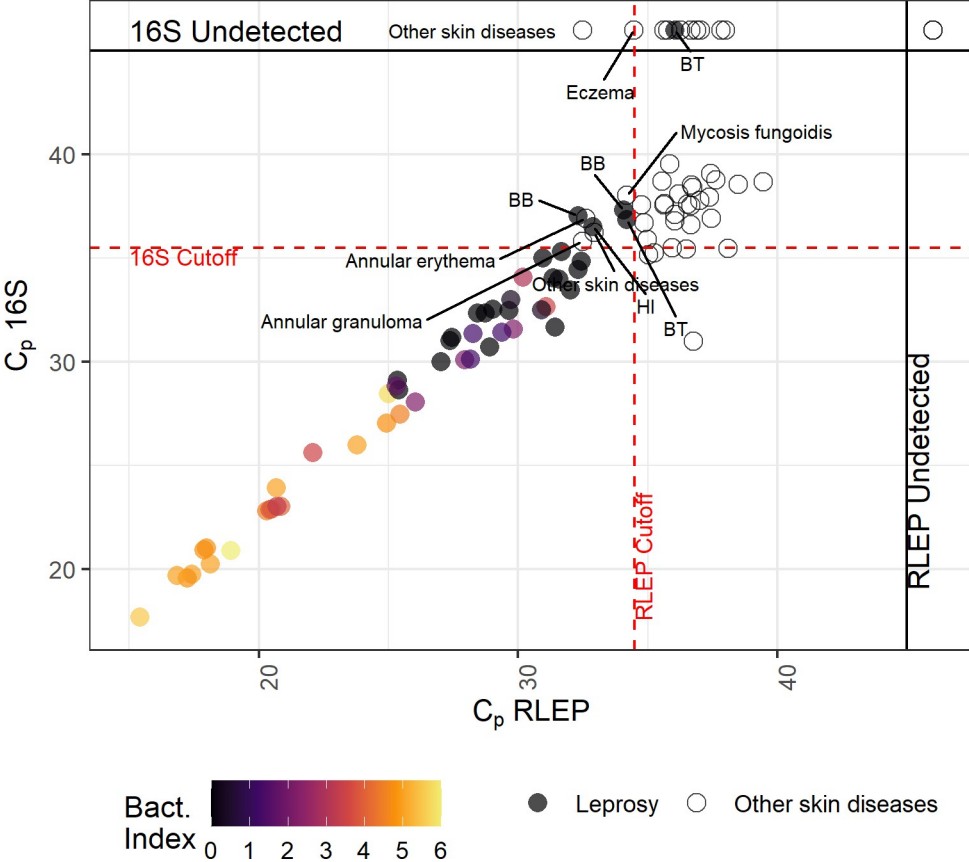

**Fig 5. Distribution of C_p values obtained for the training panel.** Each point represents a different sample (mean C_ps of a technical duplicate). Filled circles represent leprosy samples and open dots represent negative samples, as defined by the clinical assessment. Points aligned to the top and right margins indicate samples in which 16S rRNA or RLEP, respectively, were not detected within 45 cycles. Bacterial index is shown as a color gradient (samples for which bacterial index information was not available are filled in gray). Dotted red lines indicate the cutoff values from Table 3. Equivocal or misclassified samples are annotated with the operational classification (false negatives) or with the diagnosis for clinic-negative samples.

The 50 samples were classified according to our algorithm, resulting in 33 correctly classified as positive and 11 correctly classified as negative. Of the four samples classified as equivocal, two were negative for the reference method and one was positive. The sensitivity, specificity and accuracy calculated for this sample set were 97.1%, 100% and 98%, respectively.

Taken together, these results demonstrate that the multiplexed reaction for *M. leprae* RLEP and 16S rRNA is able to classify samples precisely, by combining the strengths of each molecular target and improving on their use in isolation. This setup is also able to flag samples with low bacilli burden for further monitoring, avoiding unnecessary treatment of uninfected patients and proper follow up of *M. leprae* carriers.

## Discussion

Leprosy is a chronic infectious disease with a wide range of clinical forms, each distinguished by immunological and histopathological features. Leprosy can be tuberculoid, which is a localized form exhibiting few or no bacteria, or lepromatous, which is a systemic form with high loads of mycobacteria. Among the tuberculoid patients, there is a range of skin granulomatous diseases phenotypically comparable to leprosy [5].

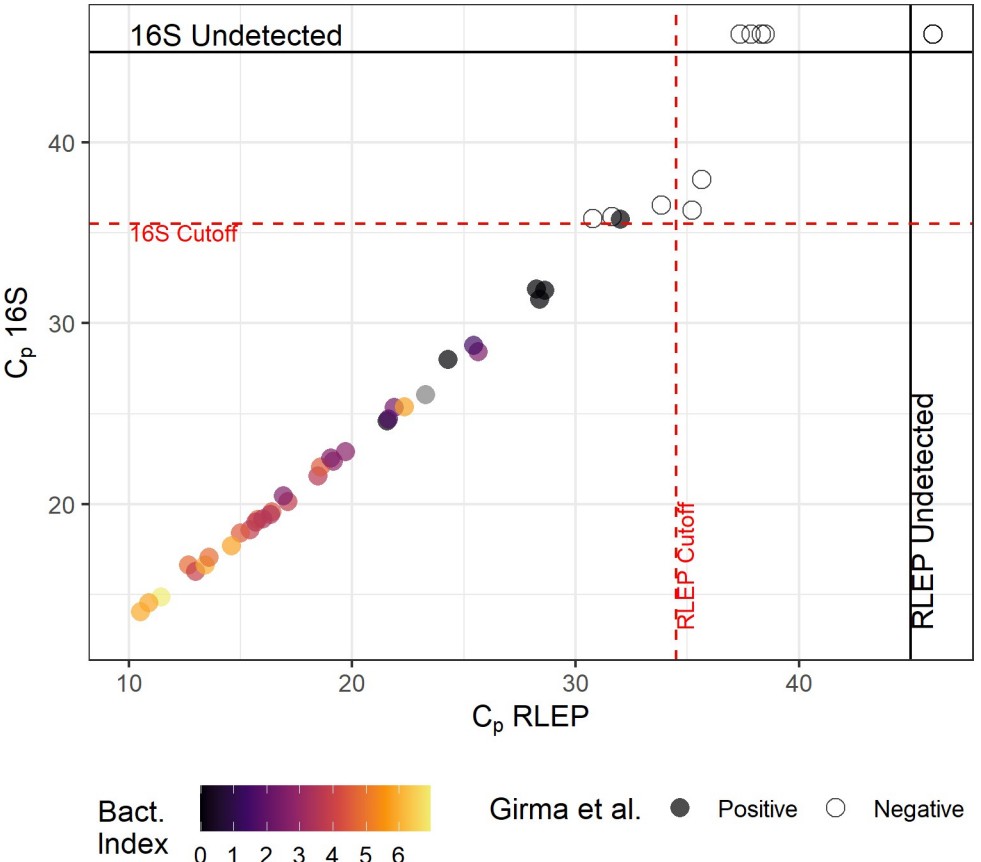

**Fig 6. Validation of parameters with an independent sample panel.** Samples previously characterized by Girma et al [32] were subjected to the new qPCR described in the present study. Each point represents a different sample. Filled circles represent leprosy samples and open dots represent negative samples. Points aligned to the top and right margins indicate samples in which 16S rRNA or RLEP, respectively, were not detected within 45 cycles. Bacterial index is shown as a color gradient (samples for which bacterial index information was not available are filled in gray). Dotted red lines indicate the cutoff values from Table 3.

The use of PCR for leprosy diagnosis has been extensively tested [4,16,35–46]. However, limitations towards the experimental designs for some published studies were identified. We observed that most studies: (i) test only samples from leprosy patients, creating difficulties in determining some diagnostic parameters such as specificity; (ii) were performed on small sample sizes; and (iii) do not have independent validation on the same assay or an evaluation of the same protocol in different centers. Furthermore, no studies have used reagents produced under good manufacturing practices (GMP), a set of guidelines that allow for traceability and batch-to-batch reproducibility of characteristics such as physical parameters and performance of the reagents [47].

In this study, we solved some of these issues by (i) developing and validating an assay based on the two most tested targets in the literature with better accuracy so far [7,8,43,48] (ii) following guidelines for validation of diagnostic tests [47,49,50], and (iii) using GMP grade reagents. We were also able to include a reaction for the detection of human 18S rRNA gene in the sample, to assess the quality of DNA extraction and reagent performance in the same reaction as the *M. leprae* determination occurs.

RLEP and 16S rRNA are the most frequent markers used in leprosy studies, displaying PCR sensitivity values up to 80% for each target. However, it is important to note that the sensitivity

of targets varied between sample types, clinical settings, and also between studies of the same authors [8,9]. Tatipally et al. [9] showed that using more than one marker in a multiplex format of conventional endpoint PCR yields significantly higher PCR positivity.

In the current study, a multiplex qPCR assay simultaneously amplifies two specific *M. leprae* targets (16S rRNA and RLEP), and the mammalian 18S rRNA gene as internal reaction control. The assay validation comprised analytical performance, diagnostic sensitivity and specificity, as well as reproducibility and repeatability. Development of multiplex qPCR assays provides a greater challenge than designing singleplex assays because it often requires extensive optimization as primer-dimers and non-specific interactions may interfere with amplification of the desired targets. Additionally, it is important that the amplification of two or more targets does not preferentially amplify one of the targets [51,52]. Combining multiple primers and probes did not affect the efficiency of the triplex qPCR in comparison to the corresponding singleplex reactions used in Martinez et al. [16]). They evaluated the independent detection of 16S rRNA and RLEP using the same primers and probes and obtained 0.51 and 0.91 for sensitivity and 0.73 and 1 for specificity, respectively. Barbieri et al. [4] also used the same 16S rRNA target to evaluate paucibacillary leprosy samples and obtained 0.57 for sensitivity and 0.91 for specificity. Here, we evaluated a panel with 53 leprosy and 44 non-leprosy patient samples, and later a different sample panel (50 patient samples) and achieved high sensitivity (> 90%) and specificity (100%) for both panels tested.

However, we understand that the small number of paucibacillary (PB) individuals in our study is a limitation. In fact, the greatest importance of using qPCR as a complementary diagnosis is precisely for PB samples. Generally, PB patients exhibit low (or zero) bacterial index and the histopathology examination does not distinguish them from other dermatoses. Therefore, these are the cases where clinical evaluation alone might not be able to determine the diagnosis, and where a qPCR confirmation becomes more important. However, due to the scarcity of bacterial DNA in these samples, it is known that the detection of *M. leprae* in PB patients by real-time PCR is difficult [4].

The reactions we developed in this study predict the equivocal classification of early-stage infections based on the finding from Martinez et al. [16], who showed that RLEP displays higher sensitivity than 16S rRNA whereas the ribosomal gene displays higher specificity. Thus, samples lacking 16S rRNA amplification but with RLEP amplification with a $C_p$ lower than the threshold are suggested to be re-analyzed.

In general, our data (Fig 5) show a correlation between BI and $C_p$ values. Biopsies from patients with higher BI values were deemed positive for bacteria earlier in the amplification cycle, as seen by the lower Cp values and high copy numbers of bacilli.

The "analytical sensitivity" or "limit of detection" of an assay is defined as the ability of the assay to detect very low concentrations of a given substance in a biological specimen [47]. The result of the limit of detection ($LoD_{95\%}$) determination when tested on a purified *M. leprae* sample indicated a higher sensitivity for RLEP (4.6 fg of DNA/reaction, equivalent to approximately 1.3 *M. leprae* genomes) versus 16S rRNA (450 fg of DNA/reaction, approximately 126 *M. leprae* genomes). This difference in sensitivity was expected since the 16S rRNA is a single copy gene [29] and the RLEP presents an average of 36 copies per genome [26].

Applicability in a reference laboratory setting was also considered during the development of these reactions. Intra and inter-operator variability were low, ensuring consistent results in routine testing (S3 Table). Moreover, reagents remained stable for at least five months, allowing for adequate stock maintenance (Fig 4).

Leprosy is a silent disease with a very long incubation time. Currently, transmission can only be halted if patients obtain early diagnosis. High-risk individuals, which are the patients' close contacts, should be traced and treated whenever leprosy is detected. Recently, it has been

suggested that novel policies towards this group of contacts such as immuno- and chemoprophylaxis are effective to help control the disease burden [15,53,54]. These approaches provide a screening of the high-risk population that, coupled with a pharmacological or immunological intervention, has been suggested to decrease disease incidence.

In some situations, clinical diagnosis needs the accuracy of a laboratory analysis, and qPCR is a reliable technique to enable diagnostic confirmation [10]. Indeed, we confirmed that the availability of molecular tests can be very helpful in diagnosing patients during contact monitoring [55]. When contacts present a leprosy-like lesion, a positive PCR has resulted in a leprosy diagnosis with 50% sensitivity and 94% specificity [55]. Other indirect methods based on simultaneous detection of host humoral as well as cellular immune responses directed against the bacteria are also promising new diagnostic tools. Recently, lateral flow assays (LFA), combining detection of mycobacterial components and host proteins, proved to be specific and sensitive [56–62]. The signature detected by this platform identified 86% of the leprosy patients, with a specificity of 90% (AUC: 0.93, p < 0.0001) [60]. Thus, a multicentric study comparing different available methods such as qPCR and LFA is still necessary. It is noteworthy that our data showed accuracy, sensitivity, and specificity values quite similar to LFA.

We believe that the diagnosis of tropical and neglected diseases needs molecular-based methods such as PCR, especially due to the robustness and capillarity of the technique in clinical analysis laboratories worldwide. Towards that future, we present a real-time quantitative PCR produced with GMP reagents that adheres to all quality control specifications, allowing batch-to-batch performance reproducibility and repeatability, and that can be used in research and clinical laboratories with reasonable infrastructure in endemic countries. Finally, we envision the multiplex qPCR assay developed adapted to more affordable, rapid, point-of-care tests to be used in low-resourced settings, enabling on-site early and specific diagnosis of leprosy, hopefully helping disease control.

## Supporting information

**S1 Text. Sequences of the synthetic DNA template control.**
(DOCX)

**S1 Fig. Diagnostic performance of the new multiplex qPCR.** Different combinations of cutoff values for 16S rRNA (panels) and RLEP (color scale) were tested on a patient panel (n = 97). For each combination of cutoff values, the sensitivity and specificity were calculated and plotted as ROC curves. Here, only $C_p$ cutoff values for 16S rRNA between 35 and 36.5 are shown. The combinations resulting in a specificity of 1 and the highest sensitivity for each condition are annotated.
(TIFF)

**S1 Table. Validation multiplex real-time qPCR assay study results.**
(XLSX)

**S2 Table. Reproducibility and Repeatability results from synthetic DNA.**
(XLSX)

**S3 Table. Precision measurement for repeatability and reproducibility.**
(XLSX)

**S4 Table. List of $C_p$ cutoff value combinations with associated sensitivity and specificity scores.**
(XLSX)

**S5 Table. Sensitivity, Specificity, PPV, NPV and accuracy of the multiplex qPCR reaction calculated for Multibacillary and paucibacillary samples.**
(XLSX)

**S6 Table. Individual $C_p$ values for targets included in the multiplex real-time qPCR assay (16S rRNA/RLEP/18S rRNA), sociodemographic, and laboratory variables for patient samples included in this study.**
(XLSX)

## Acknowledgments

The authors are grateful to the entire team of dermatologists, nurses, and technicians that collaborate at the Souza Araújo Clinic from the Leprosy Laboratory at the Oswaldo Cruz Institute. The authors are also grateful for the excellent technical assistance by Aline Burda Farias, Nilson José Fidêncio and Sylvia Mara Bohn at IBMP. We thank the Laboratório de Pesquisa em Leishmaniose (IOC-FIOCRUZ) and Laboratório de Biologia Molecular Aplicada a Micobactérias (IOC-FIOCRUZ) for donating the DNA from mycobacterial samples used in the analytical specificity study.

## Author Contributions

**Conceptualization:** Alexandre Dias Tavares Costa, Milton Ozório Moraes.

**Data curation:** Fernanda Saloum de Neves Manta, Thiago Jacomasso, Suelen Justo Maria Moreira, Charlotte Avanzi, Thyago Leal-Calvo.

**Formal analysis:** Fernanda Saloum de Neves Manta, Thiago Jacomasso, Charlotte Avanzi, Thyago Leal-Calvo.

**Funding acquisition:** Marco Aurelio Krieger, Milton Ozório Moraes.

**Investigation:** Fernanda Saloum de Neves Manta, Thiago Jacomasso, Rita de Cássia Pontello Rampazzo, Suelen Justo Maria Moreira, Najua M. Zahra, Charlotte Avanzi, Thyago Leal-Calvo, Sidra Ezidio Gonçalves Vasconcellos.

**Methodology:** Fernanda Saloum de Neves Manta, Thiago Jacomasso, Rita de Cássia Pontello Rampazzo, Marcelo Ribeiro-Alves, Alexandre Dias Tavares Costa, Milton Ozório Moraes.

**Project administration:** Alexandre Dias Tavares Costa.

**Resources:** Phillip Suffys, Marco Aurelio Krieger, Milton Ozório Moraes.

**Software:** Thiago Jacomasso.

**Supervision:** Stewart T. Cole, Phillip Suffys, Alexandre Dias Tavares Costa, Milton Ozório Moraes.

**Validation:** Fernanda Saloum de Neves Manta, Najua M. Zahra.

**Visualization:** Thiago Jacomasso.

**Writing – original draft:** Fernanda Saloum de Neves Manta, Thiago Jacomasso.

**Writing – review & editing:** Fernanda Saloum de Neves Manta, Thiago Jacomasso, Alexandre Dias Tavares Costa, Milton Ozório Moraes.

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
