## [Decision Letter · Decision Letter 0]

20 Nov 2021

Dear Dr. Costa,

Thank you very much for submitting your manuscript "Development and validation of a multiplex real-time qPCR assay using GMP-grade reagents for leprosy diagnosis." for consideration at PLOS Neglected Tropical Diseases. As with all papers reviewed by the journal, your manuscript was reviewed by members of the editorial board and by several independent reviewers. In light of the reviews (below this email), we would like to invite the resubmission of a significantly-revised version that takes into account the reviewers' comments. 

We cannot make any decision about publication until we have seen the revised manuscript and your response to the reviewers' comments. Your revised manuscript is also likely to be sent to reviewers for further evaluation.

Sincerely,

Linda B Adams

Associate Editor

Epco Hasker

Deputy Editor

Reviewer's Responses to Questions

**Key Review Criteria Required for Acceptance?**

**Methods**

-Are the objectives of the study clearly articulated with a clear testable hypothesis stated?

-Is the study design appropriate to address the stated objectives?

-Is the population clearly described and appropriate for the hypothesis being tested?

-Is the sample size sufficient to ensure adequate power to address the hypothesis being tested?

-Were correct statistical analysis used to support conclusions?

-Are there concerns about ethical or regulatory requirements being met?

Reviewer #1: (No Response)

Reviewer #2: Yes 

(Please note that all comments are shown together in the section "Summary and General Comments"

**Results**

-Does the analysis presented match the analysis plan?

-Are the results clearly and completely presented?

-Are the figures (Tables, Images) of sufficient quality for clarity?

Reviewer #1: (No Response)

Reviewer #2: Yes 

(Please note that all comments are shown together in the section "Summary and General Comments"

**Conclusions**

-Are the conclusions supported by the data presented?

-Are the limitations of analysis clearly described?

-Do the authors discuss how these data can be helpful to advance our understanding of the topic under study?

-Is public health relevance addressed?

Reviewer #1: (No Response)

Reviewer #2: Yes 

(Please note that all comments are shown together in the section "Summary and General Comments"

**Editorial and Data Presentation Modifications?**

Reviewer #1: (No Response)

Reviewer #2: Manuscript describes a new assay which has been extensively evaluated and meticulously tested. 

However, the manuscript is very lengthy at present. Several figures can be shifted to Supplementary Information if considered suitable for maintaining a continuity of the important content, which interests most of the readers.

**Summary and General Comments**

Reviewer #1: The article by Alexandre DT Costa et al. describes the development of multiplex real-time qPCR for the early diagnosis of leprosy. The evaluation of the diagnostic test using real time qPCR with clinical specimens of leprosy as well as other dermatological specimens has shown good performance in terms of sensitivity and specificity.

Major concern is Figure 6, which shows the distribution of Cp values of multiplex real time qPCR. It is questionable whether multiplex qPCR (using both RLEP and 16sRNA) rather than detection of RLEP alone, is necessary for leprosy diagnosis especially in a clinical setting. For the detection of M. leprae, RLEP is generally more sensitive than 16S rRNA in a PCR reaction (well accepted observation), but 16S rRNA may be show higher specificity. To prove that multiplex qPCR (using both RLEP and 16sRNA) is more efficient, direct comparison and statistical evaluation of real time qPCR using RLEP alone and both RLEP and 16sRNA is needed.

The accuracy, reproducibility as well as the stability of the multiplex qPCR should be evaluated using the DNA samples from the biopsies in addition to the synthetic DNA template.

Minor comments:

Table 1 shows the characteristics of the patients including both leprosy and other dermatological diseases. Criteria for the selection of the 2nd panel has to be described in the Table or Methods. PN describes Pure Neural cases of leprosy but none seem to be involved in the study.

Reviewer #2: Reviewer’s comments on the manuscript entitled " Development and validation of a multiplex real-time qPCR assay using GMP-grade reagents for leprosy diagnosis.” 

Brief summary: The authors have developed a multiplex qPCR targeting 16S and RLEP genomic regions of M. leprae and have included human 18S rRNA sequences as an inbuilt control of successful DNA extraction from patient clinical samples. They have validated their assay in test and validation set of samples and have reported excellent sensitivity and specificity. This is a very important development and work is carried out very well. A few query are shown below. In addition, a few changes may be incorporated to improve the manuscript further.

Major comments:

1. As identified by the authors, there are fewer samples of paucibacillary patients. As the sensitivity of a test varies significantly based on the number of paucibacillary samples included in the analysis, the authors should present their findings about the test performance parameters separately for the paucibacillary and multibacillary samples. 

2. Positive Predictive Value (PPV) and Negative Predictive Values (PPV) of the qPCR assay can be mentioned while writing the Sensitivity, Specificity and Accuracy, preferably separately for the paucibacillary and multibacillary cases .

3. If there was a selection criteria of the patients (whether consecutive samples, or from a specified duration), please mention it after the sentence in line# 160.

4. The Table 1 (on page 10), can be updated with the qPCR results as one more column, which will enable the reader to identify the qPCR positivity trends for the PB and MB cases; and its correlation with the Bacterial Index (written as “Bacterial load” in Table 1). Indeed, “Bacterial load” can be replaced with B.I. to be more accurate.

5. Line 398: Regarding the samples in the “Equivocal” quadrant, has there been a re-testing of the same DNA samples? This is important since as a new sample is generally difficult to obtain in most settings.

6. Lines 473-477: The sentence is not clear “

Combining multiple primers and probes did not affect the efficiency of the triplex qPCR in comparison to the corresponding singleplex reactions used in Martinez et al. (16), who evaluated the independent detection of 16S and RLEP using the same primers and probes and obtained 0.91 and 0.51 for sensitivity and 0.73 and 1 for specificity, respectively.

Does it mean, the sensitivity of 16S rRNA based detection was higher (0.91) than that of RLEP (0.51), or is it listed in reverse order? 

Authors may check this statement again and reword/correct it as necessary. 

7. Are these primers described for 16S rRNA specific for M. leprae, and would they give any results with M. lepromatosis also? Since M, lepromatosis is also a causative agent in a significant number of cases, specially in Mexico and other countries, it would be good that this information is included in the paper. Authors can state this with experimental data or can add a small paragraph with the bioinformatic analysis of the corresponding sequences regarding the level of matching of the primers with M. lepromatosis sequences. 

8. This article can be shortened and a lot of material can be shifted to supplementary Information.

9. Line 293-294: Please mention how the “450 fg of DNA (ca. 126 M. leprae genomes) is converted? Is it based on the microscopy based estimation of the bacterial counts or based on the Qubit/Nanodrop based estimation of the DNA quantity converted into bacterial DNA copy number? What was the source of the DNA sample and how contribution of the host DNA nullified in the DNA preparation, if the standard DNA was from an infected animal tissue? 

10. Line 229: “1.83 to 1.83 x 10E+07 copies/reaction”. Can it be represented as “1.83X10E+0 to 1.83 x 10E+7) to imply the intended meaning more clearly? 

11. Line 274-276: Cp (Crossing point) values for each target were set to intercept the positive controls and avoid the negative ones, being established as follows: 0.2 for RLEP, 0.15 for 16S rRNA, and 0.16 for 18S rRNA”. Are these values static between different experiment or can vary between different experiments? Is the background adjustment necessary for each experiment? 

Minor comments:

12. One more round of careful proofreading of the manuscript will be helpful, to find out some minor typos such as “16SrRNA” “18SrRNA” (space).

13. A few corrections in language are needed, for example: the sentence “Bacterial load is show as ….” . Likewise, in line 310 “ associated with skin diseases such Leishmania …:

14. Line 301 (tom and Bottom): Top and bottom.

PLOS authors have the option to publish the peer review history of their article (what does this mean?). If published, this will include your full peer review and any attached files.

Reviewer #1: No

Reviewer #2: No
---

## [Editor Report · Decision Letter 1]

26 Jan 2022

Dear Dr. Costa,

We are pleased to inform you that your manuscript 'Development and validation of a multiplex real-time qPCR assay using GMP-grade reagents for leprosy diagnosis.' has been provisionally accepted for publication in PLOS Neglected Tropical Diseases.

Best regards,

Linda B Adams

Associate Editor

Epco Hasker

Deputy Editor

---

## [Editor Report · Acceptance letter]

16 Feb 2022

Dear Dr. Costa,

We are delighted to inform you that your manuscript, "Development and validation of a multiplex real-time qPCR assay using GMP-grade reagents for leprosy diagnosis.," has been formally accepted for publication in PLOS Neglected Tropical Diseases.

Best regards,

Shaden Kamhawi

co-Editor-in-Chief

Paul Brindley

co-Editor-in-Chief
